# Prevalence and burden of bronchiectasis in a lung cancer screening program

Maria Sanchez-Carpintero Abad[1]*, Pablo Sanchez-Salcedo[2], Juan P. de-Torres[3], Ana B. Alcaide[3], Luis M. Seijo[4], Jesus Pueyo[5], Gorka Bastarrika[5], Javier J. Zulueta[3‡], Arantza Campo[3‡]

1 Pulmonary Department, Hospital Universitario Infanta Elena, Valdemoro, Madrid, Spain, 2 Complejo Hospitalario de Navarra, Pamplona, Spain, 3 Pulmonary Department, Clínica Universidad de Navarra, Pamplona, Spain, 4 Pulmonary Department, Clínica Universidad de Navarra, Madrid, Spain, 5 Radiology Department, Clínica Universidad de Navarra, Pamplona, Spain

‡ These authors are joint senior authors on this work.
* carpin.maria@gmail.com

## Abstract

### Introduction

The prevalence of bronchiectasis in the general population and in individuals undergoing lung cancer screening with low dose computed tomography (LDCT) is unknown. The aim of this study is to estimate the prevalence and impact of bronchiectasis in a screening lung cancer program.

### Methods

3028 individuals participating in an international multicenter lung cancer screening consortium (I-ELCAP) were selected from 2000 to 2012. Patients with bronchiectasis on baseline CT were identified and compared to selected controls. Detection of nodules, need for additional studies and incidence of cancer were analyzed over the follow-up period.

### Results

The prevalence of bronchiectasis was 11.6%(354/3028). On the baseline LDCT, the number of subjects with nodules identified was 189(53.4%) in patients with bronchiectasis compared to 63(17.8%) in controls (p<0.001). The occurrence of false positives was higher in subjects with bronchiectasis (26%vs17%;p = 0.003). During follow-up, new nodules were more common among subjects with bronchiectasis (17%vs.12%; p = 0.008). The total number of false positives during follow-up was 29(17.06%) for patients with bronchiectasis vs. 88(12.17%) for controls (p = 0.008).The incidence rate of lung cancer during follow-up was 6.8/1000 and 5.1/1000 person-years for each group respectively (p = 0.62)

### Conclusions

Bronchiectasis are common among current and former smokers undergoing lung cancer screening with LDCT. The presence of bronchiectasis is associated with greater incidence

Data Availability Statement: The data underlying the results presented in the study are available

from https://doi.org/10.6084/m9.figshare.11788836.v1.

**Funding:** P-IELCAP was supported (in part) by a grant (RD12/0036/0062 and RD12/0036/0040) from Red Temática de Investigación Cooperativa en Cáncer (RTICC), Instituto de Salud Carlos III (ISCIII), Spanish Ministry of Economy and Competitiveness & amp; European Regional Development Fund (ERDF) "Una manera de hacer Europa". Spanish Ministry of Health FIS Projects: PI04/2404, PI04/2128, PI07/0792, PI10/01652, PI10/00166, PI11/01626, PI13/00806, and CIBERES.

**Competing interests:** The authors have declared that no competing interests exist.

of new nodules and false positives on baseline and follow-up screening rounds. This leads to an increase need of diagnostic tests, although the lung cancer occurrence is not different.

## Introduction

Lung cancer is the leading cause of cancer-related mortality worldwide. Lung cancer screening using low dose computed tomography (LDCT) has been shown to be effective, albeit with concerns regarding potential harms due to false positives. Benign nodules are common and may prompt unnecessary tests, patient anxiety and increased costs. Improving the selection of high risk individuals whom will benefit from lung cancer screening will minimize harms and result in a lower risk of overdiagnosis [1,2] Patients with bronchiectasis may have a higher risk of false positives due to infections and inflammation.

Bronchiectasis frequently causes respiratory symptoms, loss of pulmonary function, impairment in quality of life and early mortality, although the prevalence in the general population is unknown [3–5]. The increasing use of computed tomography, especially for lung cancer screening, is impacting on the diagnosis of bronchiectasis allowing for better identification and characterization of the disease [6–10].

The aim of this study is to assess the prevalence of bronchiectasis in participants of a lung cancer screening program, and to evaluate the burden of this condition in terms of LDCT false positives and need for additional studies [9,11,12].

## Methods

### Subjects and cancer screening protocols

Subjects were selected from the Pamplona (Clínica Universidad de Navarra, Spain) subcohort of the International Early Lung Cancer Action Program (I-ELCAP) between 107 2000 to 2012 and followed until 2014 (n = 3028) [13].

The study included adults of ≥40 years-old, with a smoking history of ≥10 pack-years, without any cancer within5 years prior to entry, and without lung cancer symptoms. The Ethics Committee of the University of Navarra approved the study protocol (project028/2012 mod 2, "I-ELCAP-FAMRI") and all subjects signed an informed consent.

The lung cancer screening protocol has been described previously [14]. Briefly, all the patients had at least one LDCT performed in a single breath-hold at end-inspiration with a multidetector CT scanner (Somaton Sensation 64) at low-radiation-dose settings (120kVp, 20–40 mAs). All CT scans were reconstructed with 1.25-mm slice thickness and 1-mm intervals using a high spatial frequency reconstruction algorithm. The test or treatments indicated as part of the protocol have been scheduled according to the criteria of the attending physician following the protocol.

On baseline LDCT four outcomes were possible: 1) no pulmonary nodules observed (annual LDCT recommended); 2) small pulmonary nodules not requiring additional tests (annual LDCT recommended); 3) pulmonary nodules that require an additional follow-up LDCT, antibiotics, biopsy or a positive emission tomography (PET) before the next annual follow-up, and have not turned out to be cancer (defined as a false positive); and 4) patients with nodules that have required additional tests and finally were cancer (true positives). The protocol establishes the criteria of baseline LDCTs with a new nodule that require additional tests.

In annual screenings there are also 4 possibilities: 1) LDCT without new nodules or growth of the previously identified nodules (annual LDCT recommended); 2) new nodules or growth

of a previously observed nodule that do not meet criteria for further tests (annual LDCT recommended); 3) new nodules or obvious growth of previously observed nodules requiring additional tests (follow-up LDCT, PET or biopsy) or treatment (antibiotics) without a final diagnosis of cancer (false positives); and 4) new nodules and/or growth of previous ones who, after additional tests, have turned out to be cancer (true positives). The protocol establishes the criteria for annual LDCTs with a suspicious growth rate of malignancy that require additional tests.

## Identification of patients with bronchiectasis and without bronchiectasis

Out of 3028 participants, 354 subjects had bronchiectasis on baseline LDCT confirmed with a double reading by a radiologist and a pulmonologist (concordance analysis 0.95, 95% CI: 0.86–0.98). The severity and extension of bronchiectasis were scored using the Bhalla score [10]. (Table 1). From the same cohort 354 controls without bronchiectasis were matched by sex, smoker or ex-smoker status and range of age and tobacco consumption in pack-years.

Characterization of patients was performed by demographics, clinical records, pulmonary function tests (PFTs) and LDCT findings. PFTs (Vmax 22, Sensormedics Corps, Yorba Linda, California) were done following ATS/ERS 2005 guidelines [15]. Airflow obstruction is defined by an FEV1/FVC<0.70. The prevalence and incidence of pulmonary nodules, the need for additional tests, and the occurrence of cancer were analyzed at baseline and over the follow-up period. The presence of coronary calcium and emphysema was detected visually.

## Data analysis

Quantitative data with normal distribution are expressed as mean ± SD and non-parametric data as median and interquartile range (IQR). Categorical data are described using absolute and relative frequencies. Comparisons between the two groups were performed using Pearson chi-square or Student test, according to variable type and distribution. Incidence rate densities were calculated with 95% confidence intervals (95%CI).

**Table 1. Bhalla score [10].**

| Category | 0 | 1 | 2 | 3 |
|---|---|---|---|---|
| Severity of bronchiectasis | Absent | Mild (luminal diameter slightly greater than diameter of adjacent blood vessel) | Moderate (lumen 2–3 times the diameter of vessel) Severe (lumen >3 times Diameter of vessel) | Severe (lumen >3 times diameter of vessel) |
| Peribronchial thickening | Absent | Mild (wall thickness equal to diameter of adjacent blood vessel) | Moderate (wall thickness greater than and up to twice the diameter of adjacent vessel) | Severe (wall thickness > 2 times the diameter of adjacent vessel) |
| Extent of bronchiectasis (no. of BP segments) | Absent | 1–5 | 6–9 | >9 |
| Extent of mucous plugging (no. of BP segments) | Absent | 1–5 | 6–9 | >9 |
| Sacculations or abscesses (no. of BP segments) | Absent | 1–5 | 6–9 | >9 |
| Generations of bronchial divisions involved (bronchiectasis/plugging) | Absent | Up to 4th generation | Up to 5th generation | Up to 6th generation and distal |
| Number of bullae | Absent | Unilateral (not >4) | Bilateral (not >4) | >4 |
| Emphysema (number of BP segments) | Absent | 1–5 | >5 | . . . |
| Collapse/consolidation | Absent | Subsegmental | Segmental/lobar | . . . |

BP: bronchopulmonary

Identification of associated factors for baseline false positives was done using binomial regression with adjustment of the variance for matched cohort data. Multilevel logistic regression analysis for mixed effects was used in order to handle matched cohort data and patients with multiple LDCTs. A two-tailed *p* value less than 0.05 was considered statistically significant. All statistical analyses were performed using Stata 15 (StataCorp, TX, USA).

## Results

The prevalence of bronchiectasis in the lung cancer screening study was 11.7% (354/3028). Despite the matching process, patients with bronchiectasis compared to controls had a mean age of 60.9±9 years vs. 59.0±9 years (p = 0.01, clinically not significant) and a smoking history of 40±2 pack-years vs. 38±20 pack-years (p = 0.30). Sex distribution between groups was similar with a male proportion of 73% and 72% (p172 = 0.80). Active smoking was less frequent in the bronchiectasis vs. control group: 218173 (61.6%) vs. 243 (68.6%) (p = 0.049).

Regarding respiratory impairment, patients with bronchiectasis had a higher frequency 175 of airflow limitation (55% vs. 45%; *p* = 0.004), lower %FEV1 (89.2% vs. 93.2%; *p* = 0.02) and a greater prevalence of emphysema (33.6 vs. 16.9%; *p*<0.001). Past history of infection, hiatal hernia or pleural disease was not different between the groups (Table 2).

Using the Bhalla score [10] (4–25 points), the median punctuation was 7 (IQR 5,13) (4–18). Patients were grouped into mild (4–10 points), moderate (11–17) and severe (18–25) disease, resulting in 307(86.7%), 46(10.2%) and 1(0.3%) patients, respectively.

### Baseline LDCT

On baseline LDCT, the number of subjects with and without bronchiectasis that hadpulmonary nodules was 189 (53,4%) and63 (17,8%), respectively (p< 0.001). Theoccurrence of false positives was also higher in patients with bronchiectasis (26%) thanin those without (17%, p = 0.003), leading to a higher proportion of follow-up LDCTs194 (24.1% vs. 15.8%, p = 0.006) and indications for antibiotics (5.93% vs. 1.69%, p = 0.003).

**Table 2. Baseline characteristics of study population by presence of bronchiectasis.**

| Characteristic | Patients with bronchiectasis | Controls without bronchiectasis | p value |
|---|---|---|---|
| | (n = 354) | (n = 354) | |
| Current smokers, n (%) | 218 (61.6) | 243 (68.6) | 0.049 |
| BMI, Kg/m$^2$ | 27.5± 4.7 | 28.0± 4.4 | 0.130 |
| Airflow obstruction, n (%) | 154 (55)* | 126 (45)** | 0.004 |
| FEV1% predicted | 89.2± 22.7* | 93.2± 18.8** | 0.020 |
| Emphysema, n (%) | 119 (33.6) | 60 (16.95) | <0.001 |
| Coronary calcium, n (%) | 130 (36.7) | 108 (30.5) | 0.080 |
| Hiatal hernia, n (%) | 25 (7.1) | 15 (4.2) | 0.104 |
| Heartburn, n (%) | 50 (14.1) | 76 (21.5) | 0.011 |
| Previous pneumonia, n (%) | 19 (5.4) | 10 (2.8) | 0.080 |
| Previous tuberculosis, n (%) | 12 (3.4) | 9 (2.5) | 0.516 |
| Asthma, n (%) | 9 (2.5) | 8 (2.3) | 0.816 |
| Previous pleural disease, n (%) | 5 (1.4) | 6 (1.7) | 0.761 |

Abbreviations: BMI; body mass index; FEV1: forced expiratory volume in 1 second. Data are expressed as mean ± SD or frequency (relative frequency).

*Spirometry available in 280 subjects.

** Spirometry available in 289 subjects.

**Table 3. Study of nodules at baseline LDCT.**

| | Bronchiectasis (n = 354) | Controls (n = 354) | p value |
|---|---|---|---|
| 1.Subjects without nodules, n (%) | 165 (46.6) | 291 (82.2) | <0.01 |
| 2. Subjects with nodules that do not meet criteria for further test, n (%) | 93 (26.2) | 2 (0.5) | <0.01 |
| 3. Subjects with nodules requiring additional tests (false positives), n (%) | 91 (26) | 59 (17) | <0.01 |
| • Additional LDCT | 85 (24.1) | 56 (15.82) | <0.01 |
| • Indication for biopsy | 3 (0.85) | 2 (0.56) | 0.654 |
| • Indication for antibiotics | 21 (5.93) | 6 (1.69) | <0.01 |
| • PET | 1 (0.28) | 6 (1.69) | 0.761 |
| 4. Lung cancer (true positives), n (%) | 5 (1.41) | 2 (0.56) | 0.254 |
| Cancer types | 2 adenocarcinoma 1 squamous 1 small cell 1 unknown | 1adenocarcinoma 1 unknown | |
| Median number of nodules (IQR) | 2 (1.3) | 1 (1.2) | <0.01 |

There were no differences in the indications for biopsy (0.85% *vs.* 0.56%, *p* = 0.654). The number of true positives in the group of bronchiectasis was 5 (1.41%) *vs.* 2 (0.56%) in the control group (*p* = 0.254).

The median number of false positive nodules was 2 (IQR:1,3) in subjects with bronchiectasis and 1 (IQR:1,2) in controls (*p* = 0.005) (Table 3).

The occurrence of false positive nodules that required additional tests was significantly associated with presence of bronchiectasis (OR 1.73; 95% CI:1.22–2.45; p = 0.002), and airflow obstruction (OR 1.94; 95% CI:1.31–2.87; p = 0.001) (Table 4).

## Annual screenings

Four-hundred forty-three (62.6%) patients returned for annual screenings (224 with bronchiectasis and 219 controls, *p* = 0.852). The median follow-up was 3.48 years (IQR:1.97,5.95) and was similar for both groups (*p* = 0.096). During follow-up, more new or growing nodules were found in patients with bronchiectasis as compared to controls (33.7% *vs.* 23.5%, *p*<0.001). The

**Table 4. Relationship of variables with false positive nodules.**

**A. Univariate analysis.**

| | Baseline | P value | Follow-up | p value |
|---|---|---|---|---|
| | OR (95 CI%) | | OR (95% CI) | |
| Bronchiectasis | 1.73 (1.22–2.45) | 0.002 | 1.55 (1.002–2.41) | 0.049 |
| Current smokers | 1.009 (0.75–1.35) | 0.949 | 1 | |
| Airflow limitation* | 1.94 (1.31–2.87) | 0.001 | 0.72 (2.51e-18-2.07e+17) | 0.987 |

**B. Multivariable analysis to assess false positives on follow-up studies**.**

| Variable | OR | Std. Err. | z | P>\|z\| | 95% CI |
|---|---|---|---|---|---|
| Airflow limitation* | 1.86 | 0.376 | 3.11 | 0.002 | 1.25–2.77 |
| Bronchiectasis | 1.57 | 0.323 | 2.21 | 0.027 | 1.05–2.435 |

Abbreviations: OR: odds ratio; 95%CI: 95% confidence interval.

*Airflow limitation at the end of the follow up period.

** Adjusted for bronchiectasis.

**Table 5. Study of nodules annual LDCT.**

|  | Bronchiectasis | Controls | P value |
|---|---|---|---|
| **Subjects** | **n = 224** | **n = 219** |  |
| • Follow-up, years, median | 3.11 (1.97–5.14) | 3.85 (1.98–6.22) | 0.09 |
| • Lung cancer, n (%) | 1 (0.44) | 5 (2.28) | 0.09 |
| • Cancer types | 1 adenocarcinoma | 3 adenocarcinoma |  |
|  |  | 1 squamous |  |
|  |  | 1 small cell |  |
| **LDCT** | **756** | **723** |  |
| • New nodules but do not require additional tests, n (%) | 105 (13.8) | 53 (7.33) | <0.01 |
| • Growth nodules but do not require additional tests, n (%) | 20 (2.65) | 24 (3.32) | 0.446 |
| • False positives, n (%) | 129 (17.06) | 88 (12.17) | <0.01 |
| O Additional LDCT | 114 (15.1) | 78 (10.8) | 0.014 |
| O Indication for biopsy | 2 (0.89) | 0 (0) | 0.393 |
| O Indication for antibiotics | 52 (6.88) 26 (3.6) <0.01 | 26 (3.6) | <0.01 |
| O PET | 10 (1.32) | 8 (1.1) | 0.705 |

number of false positives during follow-up was 128 (17.1%) for patients with bronchiectasis *vs.* 88 (12.2%) for controls ($p = 0.008$). When comparing bronchiectasis to controls, these findings led to additional LDCTs in 114 (15.1%) *vs.* 78 (10.8%, $p = 0.014$), an indication for antibiotics in 52(6.88%) *vs.* 26(3.6%, $p = 0.005$) and a PET scan in 10(1.32%) *vs.* 8(1.1%, $p = 0.705$) (Table 5).

Considering a total follow-up of 1861.52 patient-years, the incidence rate of new nodules and false positives was calculated for each group. Related to controls, the incidence rate ratio (IRR) in patients with bronchiectasis was 2.24 (95% CI:1.59–3.18; $p<0.001$) for new nodules and 1.66 (95%CI:1.25–2.20; $p<0.001$) for false positives. On univariate analysis, the occurrence of false positives on annual LDCTs was associated with presence of bronchiectasis (OR 1.55; 95%CI:1.002–2.41; $p = 0.049$) (Table 4A). On a multivariable analysis of the most relevant variables, bronchiectasis was independently associated with false positives on annual LDCTs (OR 1.57; 95%CI1.05–2.35; $p = 0.027$) (Table 4B).

## Cancer

Biopsy was recommended in 23 patients and finally performed in 20: 11 with bronchiectasis (8 after baseline LDCT, 3 during follow-up) and 9 controls (4 after baseline LDCT, 5 during follow-up). Thirteen had a diagnosis of cancer, (6 and 7 respectively, p = 0.77). The incidence rate for cancer diagnosis during follow up for patients with and without bronchiectasis was 6.8 and 5.1/1000 person-years, respectively (p = 0.62). The difference in proportion for lung cancer in baseline study was 0.84% (95%CI: -0.061% to 0.23%) p = 0.256. In annual CTs was -1.84% (95%CI -0.4% to 0.32%) p = 0.094.

## Discussion

### Prevalence of bronchiectasis

The prevalence of bronchiectasis in smokers undergoing lung cancer screening was high and had an impact on the need for additional tests, but not on the incidence of cancer. The prevalence of bronchiectasis was 11.7%, higher than other studies in adult or older population, which are between 4.2 and 1106 /100000 inhabitants in different countries and times [16–21]. This wide variability could be due to different diagnostic techniques and criteria, age range

considered or past exposure to infections. The greater age and tobacco exposure in our study may also be important factors [7].Other studies on lung cancer screening programs have found levels of prevalence of bronchiectasis ranging between 0.2%-16% [22–27]. The higher prevalence can influence the rate of false positive results on LDCT and consequently have an impact on cost-effectiveness.

## Burden of bronchiectasis

Most patients in our cohort had mild bronchiectasis, were asymptomatic and had no previous diagnosis. Although there is no validated score for daily clinical practice, the Bhalla score includes several elements that measure the extent and severity of bronchiectasis and other radiologic features associated with them [10].While other scales scarcely used in literature have some advantages in accuracy or good inter-observer variability, Bhalla score may show the severity of the disease more completely. The severity of bronchiectasis was significantly associated with age, smoking history, airflow limitation, lower FEV1, more frequent emphysema and coronary calcium, as previously seen in other studies [7,12,28–30] Airflow limitation is quite common in patients with bronchiectasis [31–33] and was found in 55% of individuals with bronchiectasis in our cohort compared to 45% of the controls. In a study of 200 patients with bronchiectasis, airway obstruction was found in 43% and the presence and severity of airflow obstruction was proportional to the severity of bronchiectasis regardless of pack-years, sex or age [3].The finding of bronchiectasis in asymptomatic individuals, albeit mild in most instances, underlines the underdiagnosis of this condition.

## Nodules and false positives at baseline LDCT

On baseline LDCT, more than a half of subjects with bronchiectasis and less than 20% of controls had nodules. In other screening studies, 13–51% of subjects have non-calcified pulmonary nodules on baseline LDCTs. In the I-ELCAP study, 13% of participants had a nodule≥5 mm [12].To our knowledge, no screening study has stratified individuals by the presence of bronchiectasis. Different criteria to define a positive LDCT in individuals with bronchiectasis may have a significant impact on costs and risk of complications due to the workup of benign nodules.

Compared to controls (16.6%), as much as 25.7% of subjects with bronchiectasis had a false positive on baseline LDCT. Other studies report 10–30% of false positives [34,35]. Our results show that having bronchiectasis increases the probability of finding nodules on baseline LDCT.

Although the number of subjects is limited, patients with bronchiectasis did not have a higher prevalence or incidence of lung cancer but underwent more additional tests. This may have an impact on the risk of complications in individuals with benign diseases and on cost-effectiveness.

## Nodules and false positives on annual LDCT

Sixty-three percent of individuals who had a baseline LDCT returned for at least one annual LDCT, which is less than in other studies in which adherence has ranged between 80%-97% [36,37], although there are other studies with lower adherence [38, 39]. One of the likely causes is that in this program the patients pay for their tests. No differences in the proportion of individuals who underwent follow-up were found between the groups.

The number of TCs with new nodules that did not require additional tests during follow-up was 15% and 7% for individuals with and without bronchiectasis, respectively. Current evidence shows that 3–10% of individuals who have an annual LDCT in the context of lung

cancer screening have new nodules [12].The proportion of false positives resulting in additional tests or treatments for a benign lesion was 17% for subjects with bronchiectasis and 12% in controls, with no differences in the number of nodules that showed growth. Bronchiectasis significantly contributes to the number of additional tests during annual rounds.

A recognized post-hoc study is carried out only for those patients older than 55 years and smokers of more than 30 pack-years according to the criteria used in other screening programs. There are 163 patients with bronchiectasis and 155 without them. There are also differences between these two groups in the baseline CT in terms of the appearance of new nodules, greater number of nodules that do not require additional tests and more false positives (in the limit of statistical significance), resulting in the indication of antibiotic. Regarding the follow-up in this population group, there are 371 LDCT with bronchiectasis and 311 controls and there are no differences between them except in the incidence of cancer. S1 Table These results may help to better characterize those subjects who should enter a lung cancer screening program or to shape in a more adapted and appropriate way the protocol to be followed in these subjects, thus avoiding the risks involved in a lung cancer screening program.

## Cancer

Perhaps bronchiectasis can also be related with cancer in its mechanism of continuous inflammation-infection. The prevalence and incidence of cancer in this sample was 1.41% (5 cancers) and 0.44% (1 cancer) in individuals with bronchiectasis *vs*. 0.56% (2 cases) and 2.28% (5 cases) in controls, respectively. No difference between groups was found, although the number of cases is small.

The I-ELCAP trial reported a cancer rate of 10% in participants with baseline nodules and 5% for participants with new incident pulmonary nodules [40]. In the Pamplona subcohort of the IELCAP consortium, from which the participants in this study were drawn, the lung cancer prevalence and incidence were 1.0% and 1.4%, respectively [41]. Our study has not been able to relate bronchiectasis to cancer but we suggest continuing in this line since it could be very useful

## Study limitations

The present study has several limitations. The data obtained cannot be generalized to the general population since the participants are smokers or former smokers, with at least 40 years and may also present a selection bias because they are volunteers concerned about their health, who are willing to pay for the tests performed. Furthermore, the IELCAP protocol requests that the radiologists check for bronchiectasis. This may favor overdiagnosis and a predominance of asymptomatic disease. However, pulmonary function tests clearly showed a higher functional impairment in these patients.

As part of usual practice there are scores of clinical severity and prognosis for bronchiectasis, but the use of radiological measurement scales is not defined. Direct or indirect signs are used in a variable way in different studies. The choice of one particular score such as the Bhalla score, that is not used in daily clinical practice, may limit comparisons with other studies. In addition, low-dose CT studies may underestimate the findings of bronchiectasis [42].

Finally, our cohort is not large enough to draw conclusions on the relationship between bronchiectasis and risk for lung cancer.

## Summary

The prevalence of bronchiectasis is high in individuals undergoing lung cancer screening with LDCT, which may have a significant impact on the rate of false positive nodules. Even with

predominantly asymptomatic disease, bronchiectasis is associated with airflow limitation, and the degree of this limitation is also correlated to with the radiographic severity of bronchiectasis. The presence of bronchiectasis increases by four-fold the number of individuals with nodules on baseline LDCT and almost doubles the mean number of nodules. In annual rounds of screening, false positives were increased by 50%in individuals with bronchiectasis. This leads to more frequent diagnostic tests without an apparently higher incidence of cancer. More studies are required to minimize the consequences of false positives in these patients.

## Supporting information

**S1 Checklist. STROBE statement—Checklist of items that should be included in reports of** *cohort studies.*
(DOC)

**S1 Table. Post-hoc study for patients older than 55 years and smokers of more than 30 pack-years.**
(DOCX)

## Author Contributions

**Conceptualization:** Maria Sanchez-Carpintero Abad, Javier J. Zulueta, Arantza Campo.

**Data curation:** Maria Sanchez-Carpintero Abad, Ana B. Alcaide, Luis M. Seijo, Jesus Pueyo, Gorka Bastarrika, Javier J. Zulueta, Arantza Campo.

**Formal analysis:** Maria Sanchez-Carpintero Abad, Arantza Campo.

**Funding acquisition:** Javier J. Zulueta.

**Investigation:** Maria Sanchez-Carpintero Abad, Javier J. Zulueta, Arantza Campo.

**Methodology:** Maria Sanchez-Carpintero Abad, Javier J. Zulueta, Arantza Campo.

**Project administration:** Maria Sanchez-Carpintero Abad, Juan P. de-Torres, Arantza Campo.

**Supervision:** Pablo Sanchez-Salcedo, Juan P. de-Torres, Luis M. Seijo, Gorka Bastarrika, Javier J. Zulueta, Arantza Campo.

**Validation:** Javier J. Zulueta, Arantza Campo.

**Visualization:** Maria Sanchez-Carpintero Abad, Arantza Campo.

**Writing – original draft:** Maria Sanchez-Carpintero Abad.

**Writing – review & editing:** Maria Sanchez-Carpintero Abad, Arantza Campo.

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
