## [Decision Letter · Decision Letter 0]

15 Jan 2020

PONE-D-19-25470

Prevalence and burden of bronchiectasis in a lung cancer screening program

PLOS ONE

Dear Dr Sánchez-Carpintero Abad,

Thank you for submitting your manuscript to PLOS ONE. After careful consideration, we feel that it has merit but does not fully meet PLOS ONE’s publication criteria as it currently stands. Therefore, we invite you to submit a revised version of the manuscript that addresses the points raised during the review process.

Please consider all required changes from the reviewers.

We would appreciate receiving your revised manuscript by Feb 29 2020 11:59PM. To enhance the reproducibility of your results, we recommend that if applicable you deposit your laboratory protocols in protocols.io, where a protocol can be assigned its own identifier (DOI) such that it can be cited independently in the future. For instructions see: http://journals.plos.org/plosone/s/submission-guidelines#loc-laboratory-protocols

We look forward to receiving your revised manuscript.

Kind regards,

Peter M.A. van Ooijen, Ph.D.

Academic Editor

PLOS ONE

Journal Requirements:

"P-IELCAP was supported (in part) by a grant (RD12/0036/0062 and RD12/0036/0040) from Red Temática de Investigación Cooperativa en Cáncer (RTICC), Instituto de Salud Carlos III (ISCIII), Spanish Ministry of Economy and Competitiveness & European Regional Development Fund (ERDF) “Una manera de hacer Europa”. Spanish Ministry of Health FIS Projects: PI04/2404, PI04/2128, PI07/0792, PI10/01652, PI10/00166, PI11/01626, PI13/00806, and CIBERES."

"The authors received no specific funding for this work"

Reviewers' comments:

Reviewer's Responses to Questions

**Comments to the Author**

1. Is the manuscript technically sound, and do the data support the conclusions?

Reviewer #1: Partly

Reviewer #2: Yes

2. Has the statistical analysis been performed appropriately and rigorously? 

Reviewer #1: I Don't Know

Reviewer #2: I Don't Know

3. Have the authors made all data underlying the findings in their manuscript fully available?

Reviewer #1: No

Reviewer #2: Yes

4. Is the manuscript presented in an intelligible fashion and written in standard English?

Reviewer #1: Yes

Reviewer #2: No

5. Review Comments to the Author

Reviewer #1: I commend the authors on the analysis of a large data set to evaluate the prevalence of bronchiectasis and related findings. In the related analysis of bronchiectasis and nodules, there are a few important adjustments/edits that would make the analysis more powerful and useful.

Major edits:

-Although there is reference to the screening protocol, it would be helpful to outline what is considered "positive."

-Current standard of care by 2D measurements, which is done in the US (Europe is using more volumetrics), includes a "positive" scan as > or = 6 mm. It appears the analysis was done with 5mm, which seems like a small difference but will alter the numbers, particularly when discussing "false positive." This paper will be referenced for "false positive" but will not be accurate for the current era if using an outdated definition of "positive." It would improve the analysis tremendously to use the updated definition of "positive."

-Care must be taken with analysis of impact of bronchiectasis within a screening program as the number of cancers is low. As part of this topic, the screening program includes those with a lower risk than those who qualify for screening in the US (starting age 40 with just 10 pack years. This can significantly impact the screening outcomes that does not make the generalizable to the currently qualifying population. If it is possible to separate out the population of those > or = 55 yrs old and pack years > or = 30 to determine if the numbers significantly change. (at a minimum it is important to acknowledge this difference from the currently approved lung screening population as publications like this get referenced to represent expectations within a lung screening program). This is important as "false positive" is something generally being overstated within lung screening programs and is getting broadly confused. This makes it even more important to be very clear about any false positive analysis. Line 308, "the number of subjects with new nodules during follow-up was 15% and 7% for individuals with and without bronchiectasis": What is considered "new nodules"? Does this mean a positive nodule of at least 5 mm? Or does this mean any nodule? The overall analysis of bronchiectasis and lung screening analysis could be more clear.

I do not see all the data from the analysis.

Reviewer #2: This is an important topic and I believe the presence of bronchiectasis does increase false positives. Most of my comments are related to clarity of writing and data presentation.

Do you have an estimate for Power to detect difference between these groups (i.e. cancer incidence)?

Line 138 do you mean to say that controls were matched on these variables?

Table 3A: recommend using parallel phrases as described in text. E.g. nodules that do not meet criteria for further tests; nodules requiring additional tests (false positives); lung cancer (true positives)

Table 4. Regression analysis to assess appearance of false positives on baseline and follow-up studies. Suggest rewording title to be more descriptive “relationship of variables with false positive nodules”

Might be helpful to describe what if any guidelines used for follow up, antibiotics or indicate that testing was done, antibiotics were prescribed, if at the discretion of treating provider or local study team (example, as in line 226 “an indication for antibiotics”)

Table 3 comes after Table 4 and has unclear text and formatting:

Why is there a “B” without an “A” label? Perhaps a formatting error?

Unclear what this means “Growth nodules but do not require additional tests” a growing nodule that didn’t undergo additional testing? That would not be typical

Line 303 – most reports of annual compliance in the US at least, are lower than what the authors report and even lower than what was seen in this study

Line 317 – not sure what conclusion you are drawing here (patients with bronchiectasis should not be screened?)

Line 325 the summary here seems to differ from what is shown in Table 3B. Is table 3A missing (showing prevalence)

Line 363 suggest adding qualifier for bronchiectasis to improve clarity, for example: “the degree of this limitation is also correlated to with the radiographic severity of bronchiectasis” to be more clear

6. PLOS authors have the option to publish the peer review history of their article (what does this mean?). If published, this will include your full peer review and any attached files.

Reviewer #1: No

Reviewer #2: No

---

## [Author Response · Author response to Decision Letter 0]

25 Feb 2020

Response to Reviewers

EDITOR:

1. The manuscript meets the style requirements of PLOS ONE, including file name requirements.

2. Funding information does not appear in the Acknowledgments section or other areas of the manuscript.

* Funding sources have been withdrawn from the manuscript and specified in the cover letter. I would like to change the financing status. The correct Funding Statement is: “The authors received specific funding for this work”.

3. Please include captions for your Supporting Information files at the end of your manuscript, and update any in-text citations to match accordingly.

* A table has been added as Suppoting Information.

Reviewer #1 

• Although there is reference to the screening protocol, it would be helpful to outline what is considered "positive."

* The definition for positive results is explained in the lines 122 and 130. Changes have been made as suggested by the 2nd reviewer so that the information is now more clear.

• Current standard of care by 2D measurements, which is done in the US (Europe is using more volumetrics), includes a "positive" scan as > or = 6 mm. It appears the analysis was done with 5mm, which seems like a small difference but will alter the numbers, particularly when discussing "false positive." This paper will be referenced for "false positive" but will not be accurate for the current era if using an outdated definition of "positive." It would improve the analysis tremendously to use the updated definition of "positive."

*It is true that the screening studies have used different measurements when considering a positive nodule:

. NLST (US): > or = 4 mm

. USPSTF (following National Comprehensive Cancer Network, US): > or = 6 mm

. LSS (from NLST): > or = 4 mm

. DANTE (Italy): > or = 6 mm

. DLCST (Denmark): > or = 5 mm

. MILD (Italy): > or = 5 mm

. NELSON (Netherlands / Belgium): volumetric measurements

. ITALUNG (Italy): > or = 5 mm

In fact, in our I-ELCAP protocol we are already using 6 mm as a cut-off point since 2014. However, this study was previously carried out (follow-up until 2014), with the cut-off point at 5mm also used in other programs. Unfortunately, and although it would be very useful, we do not have the results only in those larger than 6mm, and the indication for additional tests was done according to the current protocol at that time. The aim of this study was to compare the prevalence of nodules in patients without bronchiectasis vs. those without bronchiectasis, according to the current evaluation of nodules at the time of study.

• Care must be taken with analysis of impact of bronchiectasis within a screening program as the number of cancers is low. As part of this topic, the screening program includes those with a lower risk than those who qualify for screening in the US (starting age 40 with just 10 pack years. This can significantly impact the screening outcomes that does not make the generalizable to the currently qualifying population. If it is possible to separate out the population of those > or = 55 yrs old and pack years > or = 30 to determine if the numbers significantly change. (at a minimum it is important to acknowledge this difference from the currently approved lung screening population as publications like this get referenced to represent expectations within a lung screening program). This is important as "false positive" is something generally being overstated within lung screening programs and is getting broadly confused. This makes it even more important to be very clear about any false positive analysis. 

* We have carried out a posthoc and subgroup analysis. The results have been added on line 324 as well as a table in Supporting Information, and seem to be very similar to the whole group for baseline study. Differences are not as significant on follow-up CTs, although the size of the group is limiting the analysis.

• Line 308, "the number of subjects with new nodules during follow-up was 15% and 7% for individuals with and without bronchiectasis": What is considered "new nodules"? Does this mean a positive nodule of at least 5 mm? Or does this mean any nodule? 

*It refers to the percentage of TACS with the presence of any nodule of any size that does not meet the criteria for suspected malignancy of the I-ELCAP protocol:

“Newly identified solid or part-solid nodule < 3.0 mm or nonsolid nodule of any size”

It has been clarified in the manuscript on lines 123 and 132.

The number refers to CTs and not subjects and has been now corrected in the revised manuscript.

• The overall analysis of bronchiectasis and lung screening analysis could be more clear.

*Some changes have been made with the suggestions of reviewer 2 to make it more clear. For example, the results of the different groups have been exposed in an exclusive way to make it more evident.

• I do not see all the data from the analysis

*The database is currently available in Figshare: https://doi.org/10.6084/m9.figshare.11788836.v1

Reviewer #2

• Do you have an estimate for Power to detect difference between these groups (i.e. cancer incidence)?

*A prevalence of bronchiectasis of 16% has been described in other lung cancer screening programs. To estimate this prevalence with an accuracy of +/- 1%, an n = 1898 is required.

The sample necessary to estimate a difference in the occurrence of false positives, and therefore, the need for additional CTs or other studies, is calculated. The presence of false positives in lung cancer screening studies is 16% in the baseline study and 6% in the follow-up studies. To estimate a difference in the prevalence of the baseline study of 16% +/- 9% of false positives, a sample size of 334 patients is required on each side (power 0.80, alpha 0.05). For the incidence of false positives in follow-up CT scans of 6% +/- 7%, 303 patients are required on each side (power 0.80; alpha 0.05).

According to literature recommendations, 95%CI was given to inform better about the differences in proportions for lung cancer.

The difference in proportion for lung cancer in baseline study was 0.84% (95%CI: -0.061% to 0.23%) p= 0.256

The difference in proportion for lung cancer in annual CTs was -1.84% (95%CI -0.4% to 0.32%) p= 0.094.

*This data has been added on line 252

The sample size calculations before the study are:

The sample size to detect a 5-fold risk difference in cancer between the groups is 333 for each group, taking a baseline prevalence of 1%, power 80% and p=0.05.

The sample size to detect a 6-fold risk difference in cancer between the groups is 176 for each group, taking a baseline prevalence of 1.4%, power 80% and p=0.05.

So, the sample size is not enough to assess risk differences below 5-fold and 6-fold respectively.

• Line 138 do you mean to say that controls were matched on these variables?

*Yes, thanks for the correction. It has been introduced in this way in the manuscript being clearer.

• Table 3A: recommend using parallel phrases as described in text. E.g. nodules that do not meet criteria for further tests; nodules requiring additional tests (false positives); lung cancer (true positives).

*It is changed in the manuscript as suggested. The data has been corrected so that the results are exclusive and better understood.

• Table 4. Regression analysis to assess appearance of false positives on baseline and follow-up studies. Suggest rewording title to be more descriptive “relationship of variables with false positive nodules” 

*Thank you so much for the suggestion. It is changed in the manuscript as suggested.

• Might be helpful to describe what if any guidelines used for follow up, antibiotics or indicate that testing was done, antibiotics were prescribed, if at the discretion of treating provider or local study team (example, as in line 226 “an indication for antibiotics”).

* It is added in the manuscript that the tests or treatments have been indicated at the discretion of the attending physician, following the protocol recommendations, line 116. Thank you so much for the suggestion.

• Table 3 comes after Table 4 and has unclear text and formatting:

Why is there a “B” without an “A” label? Perhaps a formatting error?

*There is indeed an error and it is unclear. Therefore the tables now follow one another with numbers up to 5, without breakdown by A and B, except for Table 4.

• Unclear what this means “Growth nodules but do not require additional tests” a growing nodule that didn’t undergo additional testing? That would not be typical

*The protocol specifies the growth criteria that it considers are associated with the indication of additional tests:

“Percentage change consistent with malignancy = (diameter at time 2 - diameter at time 1)/ diameter at time 1

a) if nodule < 6 mm in diameter, % change > 50% growth

b) nodules 6-9 mm, % change > 30% growth

c) nodules > 10 mm, % change > 20% growth”

It has been clarified in the manuscript on lines 123 and 132.

• Line 303 – most reports of annual compliance in the US at least, are lower than what the authors report and even lower than what was seen in this study

*Indeed there are other studies with lower adherence and this data has been included in the manuscript, line 304. Thanks for the suggestion.

• Line 317 – not sure what conclusion you are drawing here (patients with bronchiectasis should not be screened?) 

* Perhaps patients with bronchiectasis that have been seen to need more follow-up and also have worse lung function, should be followed by a pulmonologist in clinical consultations. More studies would be needed to assess whether these patients should follow a different protocol in lung cancer screening given the findings.

• Line 325 the summary here seems to differ from what is shown in Table 3B. Is table 3A missing (showing prevalence) 

*The number have been revised and corrected as indicated.

• Line 363 suggest adding qualifier for bronchiectasis to improve clarity, for example: “the degree of this limitation is also correlated to with the radiographic severity of bronchiectasis” to be more clear.

*We appreciate the suggestion. It is changed in the revised manuscript. 

Thank you very much

---

## [Decision Letter · Decision Letter 1]

19 Mar 2020

Prevalence and burden of bronchiectasis in a lung cancer screening program

PONE-D-19-25470R1

Dear Dr. Sánchez-Carpintero Abad,

We are pleased to inform you that your manuscript has been judged scientifically suitable for publication and will be formally accepted for publication once it complies with all outstanding technical requirements.

With kind regards,

Peter M.A. van Ooijen, Ph.D.

Academic Editor

PLOS ONE

Additional Editor Comments (optional):

Reviewers' comments:

Reviewer's Responses to Questions

**Comments to the Author**

1. If the authors have adequately addressed your comments raised in a previous round of review and you feel that this manuscript is now acceptable for publication, you may indicate that here to bypass the “Comments to the Author” section, enter your conflict of interest statement in the “Confidential to Editor” section, and submit your "Accept" recommendation.

Reviewer #1: All comments have been addressed

2. Is the manuscript technically sound, and do the data support the conclusions?

Reviewer #1: (No Response)

3. Has the statistical analysis been performed appropriately and rigorously? 

Reviewer #1: (No Response)

4. Have the authors made all data underlying the findings in their manuscript fully available?

Reviewer #1: (No Response)

5. Is the manuscript presented in an intelligible fashion and written in standard English?

Reviewer #1: (No Response)

6. Review Comments to the Author

Reviewer #1: line 325 appears to have the incidence and prevalence values reversed.

Author response in reference to question about what is considered "new nodules" in initial line 308 provides quote "Newly identified solid or part-solid nodule < 3.0 mm or nonsolid nodule of any size." I do not see this in the text. The edit to lines 123 and 132 do not make it more clear what is considered a "new nodule". The quoted text provided in the author answer could be included if this is accurate. It would strengthen the paper to specify more explicitly how the nodules are being categorized.

7. PLOS authors have the option to publish the peer review history of their article (what does this mean?). If published, this will include your full peer review and any attached files.

Reviewer #1: No

---

## [Editor Report · Acceptance letter]

23 Mar 2020

PONE-D-19-25470R1 

Prevalence and burden of bronchiectasis in a lung cancer screening program 

Dear Dr. Sanchez-Carpintero Abad:

I am pleased to inform you that your manuscript has been deemed suitable for publication in PLOS ONE. Congratulations! Your manuscript is now with our production department. 

With kind regards,

on behalf of

Mr Peter M.A. van Ooijen 

Academic Editor

PLOS ONE